# Assessing Mental Health Literacy Among Undergraduate Nursing Students in South Africa

**DOI:** 10.3390/nursrep15090337

**Published:** 2025-09-15

**Authors:** Ernest Peresu, Gladys Kigozi-Male, Michelle Engelbrecht

**Affiliations:** Centre for Health Systems Research & Development, University of the Free State, Bloemfontein 9300, South Africa; kigozign@ufs.ac.za (G.K.-M.); engelmc@ufs.ac.za (M.E.)

**Keywords:** human resources for mental health, mental health, mental health disorders, mental health illness, mental health literacy, mental health providers, mental illness, psychiatric nurses, nursing students

## Abstract

**Background:** Amidst a growing shortage of human resources for mental health, concerns persist over the mental health literacy (MHL) and readiness of nursing students to fulfil their future role as non-specialist mental healthcare providers. **Methods:** A cross-sectional study was conducted to assess the MHL among nursing students enrolled in a four-year nursing programme using the Mental Health Literacy Scale (MHLS) adapted and validated for primary health care workers in the South African context. Multiple regression analysis was used to determine the factors influencing MHL among the nursing students. **Results:** Only a third of the 220 respondents (*n* = 73; 33.2%) expressed interest in specialising in mental health. A one-way ANOVA revealed statistically significant differences in the MHLS scores across year levels, F(3, 216) = 3.225, *p* = 0.023, though Tukey post hoc analysis found no significant difference between second-year students (with mental health theory only) and third-/fourth-years (with both theoretical and clinical exposure, *p* = 0.295). Year of study, family history of mental illness, and career aspirations in mental health were significant predictors of MHL. While gender initially appeared significant, a sensitivity analysis suggested that this result may not be robust due to the small number of male participants. **Conclusions:** The absence of a significant difference in MHL among nursing students across years of study suggests limitations in how the nursing training programme integrates mental health theory with clinical practice. Nursing education stakeholders should review curricula and teaching strategies to ensure that MHL is effectively reinforced throughout training.

## 1. Introduction

Mental health challenges are a major public health concern, with untreated mental disorders accounting for 13% of the total global disease burden [1]. This global challenge is further compounded by the inability of health systems worldwide to adequately respond to this growing problem. Consequently, the gap between the need for treatment for mental health disorders and its provision is substantial, with 76% to 85% of individuals with severe mental health challenges in low- and middle-income countries remaining untreated [2].

The dramatic surge in the demand for mental health services has further exacerbated challenges posed by a longstanding shortage of human resources for mental health, in particular, psychiatric nurses. It is widely acknowledged that the existing human resources for mental health are neither large enough nor sufficiently skilled and prepared to meet the needs of the population, particularly for those experiencing moderate to severe mental health conditions [1,2]. In Africa, for example, the number of specialised and general health workers dealing with mental health issues is 1.6 per 100,000 population, significantly lower than the global median of 13 per 100,000 population [1]. Under these conditions, it is unsurprising that the delivery of mental health care and the associated professional roles within it have undergone considerable change in recent years.

Drawing from the World Health Organization’s Comprehensive Mental Health Action Plan 2013–2030 [2], in recent years, South Africa has implemented a series of policies to improve mental health services. For instance, the South African National Mental Health Policy Framework and Strategic Plan 2023–2030 has been a crucial catalyst in advancing efforts to integrate mental health services into primary health care (PHC) [3]. The incorporation of mental health into PHC services, amidst severe human resources for mental health shortages in South Africa [4], implies that non-mental health specialists may have to render health services to individuals with mental illness.

In South Africa, nurses of all categories constitute the largest proportion of frontline health care professionals and are essential in expanding access to care, especially in rural areas, across the continuum of care [5]. Accordingly, there is a growing need to equip future nurses with a comprehensive, fit-for-purpose skill set that includes mental health literacy (MHL) for all clinical settings. MHL is a multifaceted concept defined as the “knowledge and beliefs about mental disorders which aid their recognition, management or prevention” [6]. To fulfil their role as future frontline mental health care providers, nursing students should have adequate MHL through exposure to mental health theory and clinical practice in their undergraduate education, sufficient for preparing them for the practical demands they will face in the field.

Globally, scholars have raised concerns that the current undergraduate nursing curriculum inadequately prepares students, largely due to its structure and limited inclusion of mental health content [7,8,9,10]. In the South African context, preparation in mental health is shaped by the leadership, decisions, strategic priorities, and regulatory role of the South African Nursing Council (SANC). The shift from ‘legacy’ nursing qualifications towards new programmes aligned with the higher education framework was first proposed in the National Strategic Plan for Nurse Education and Training and Practice, 2012/13–2016/17, with legacy programmes terminating in 2015 [11]. In line with this, SANC developed new nursing programmes to support progression and articulation within the higher education system. The National Strategic Direction for Nursing and Midwifery Education and Practice 2020/21–2025/26, introduced by the South African Department of Health, further reinforced this shift, emphasising the need to redefine the competency frameworks responsive to the country’s current and future health care needs [5]. However, these reforms have resulted in a significant reduction in mental health nursing curricular content and clinical placement experience in undergraduate programmes. Most notably, the shift has been criticised for potentially leaving nursing graduates underprepared to meet mental health care demands [12,13]. Similar concerns have also been reported in high-income countries, such as Australia, where limited MHL contributed to ongoing stigma and anxiety among undergraduate nursing students towards individuals with mental health conditions [8].

The adequacy of mental health education is not merely a curricular concern—it has direct implications for clinical outcomes and patient experiences. The consequences of nurses’ low MHL on patient outcomes have been extensively researched and documented. Numerous studies have established that nurses with limited MHL may struggle with early recognition of mental health disorders and their symptoms [8,14], leading to delays in the initiation of appropriate treatment and referrals and, consequently, poor outcomes for individuals with mental health challenges. There is also a consensus between a South African empirical study [15] and a systematic review by Hartley et al. [16], suggesting that insufficient mental health knowledge among nurses may inadvertently perpetuate stigmatising attitudes towards mental health patients and mental health care. Mental health providers’ attitudes towards and understanding of mental illness are recognised as crucial in establishing meaningful therapeutic relationships with patients experiencing mental health disorders [17].

Although concerns regarding gaps in nursing mental health education in South Africa persist, little is known about the MHL of nursing students, and, consequently, their preparedness for mental health practice remains unclear. Existing research on MHL has largely targeted qualified nurses and other health care workers [18,19], with limited focus on nursing students. The available research among South African nursing students is predominantly confined to unpublished theses [20,21], limiting its contribution to refining mental health training programmes. The current study addresses this gap by assessing the characteristics of and factors associated with MHL among undergraduate nursing students across different academic years at a South African university. Findings from this study could offer evidence-based insights important for reviewing and improving mental health education for nursing students and their readiness for mental health practice.

## 2. Methods

### 2.1. Design and Setting

A cross-sectional survey was conducted among undergraduate nursing students enrolled in a four-year comprehensive nursing programme at a university in South Africa.

### 2.2. Participants

All first- to fourth-year nursing students registered for the undergraduate nursing programme during 2023 were invited to complete the survey available in English, online and in hard copy. At this university, the classroom component of mental health theory is introduced in the second year of the programme, and further expanded in the third and fourth year, focusing on assessment, mental health promotion and the management of common psychiatric disorders. In the third and fourth year, students also gain practical experience through clinical training, known as work-integrated learning, to prepare them to be well-rounded healthcare professionals. The inclusion of first-year students in the survey provides insights into the baseline MHL and helps in identifying potential gaps in MHL prior to formal exposure to mental health theory, while the inclusion of second-, third-, and fourth-year students allows for the exploration of how the curriculum and clinical exposure might impact their MHL.

### 2.3. Data Measurement

A structured self-administered questionnaire was designed for data gathering, measuring socio-demographic variables including gender, age, year of study, perceptions about their mental health, family history of mental health conditions, knowledge of a friend with a mental health condition, personal experience with mental health illness, and whether students would consider specialising in mental health after graduation.

The General Health Questionnaire-28 (GHQ-28) [22] was used to assess nursing students’ self-perceived mental well-being and to detect individuals at risk for possible mental health disorders. This self-report screening tool is widely used in student populations, including nursing students [23,24,25,26]. The scale is grouped into four sub-scales: (1) Somatic symptoms—assessing physical symptoms related to psychological distress; (2) Anxiety and insomnia—evaluating anxiety and sleep disturbances; (3) Social dysfunction—measuring difficulties performing everyday social roles; and (4) Severe depression—to identify symptoms of severe depression. Each sub-scale consists of seven items, with responses ranging from “better than usual/not at all/more than usual/quicker than usual” to “much worse than usual/much more than usual/longer than usual”. Two primary scoring methods are used for the GHQ-28: the binary scoring method (0-0-1-1) and a Likert scoring method. This study used the latter scoring method, where responses were rated from 0 to 3, with a total possible score ranging from 0 to 84. Higher scores reflect greater psychological distress [22]. Studies among nursing students have reported satisfactory reliability, with Cronbach alpha coefficients ranging between 0.76 and 0.90 [24,26]. In the current study, the GHQ-28 had a Cronbach’s alpha of 0.95. 

MHLS was assessed using the Mental Health Literacy Scale (MHLS), originally developed for the general population by O Connor and Cassey [27], and later adapted and validated for primary health care workers in the South African context by Korhoen and colleagues [28,29]. The MHLS consists of 35 items that span six core domains conceptualised by Jorm et al. [6], including (1) recognition of mental health disorders, (2) beliefs about the causes of mental health disorders, (3) knowledge of self-help strategies, (4) understanding of professional help options, (5) attitudes that promote recognition and help seeking, and (6) knowledge of how to find mental health information. Korhoen et al.’s [29] adaptation introduced minor adjustments to the 35 items to enhance their cultural and contextual relevance, as well as ensure comprehension among respondents with diverse educational backgrounds. The scale uses two types of Likert scoring: items 1–15 are rated on a 4-point Likert scale from 1(very unlikely) to 4 (very likely), while items 16–35 are rated on a 5-point Likert scale ranging from 1 (strongly agree/definitely willing) to 5 (strongly disagree/definitely unwilling). Additionally, five items (Q10, Q12, Q15, Q20, and Q28) are reverse scored. The scale demonstrated good reliability, with a Cronbach’s alpha of 0.80, indicating satisfactory internal consistency across the instrument. In the current study, the MHLS had a Cronbach’s alpha of 0.77.

### 2.4. Data Gathering

Data were gathered from October to November 2023, a period during which students were preparing for their end-of-year examinations. Due to time constraints and concerns about overburdening participants during this critical academic period, a formal pilot study was not conducted. However, to mitigate this limitation, the study used well-established, standardised scales with demonstrated validity and reliability in similar contexts [28]. Additionally, the research team conducted an internal review of the questionnaire to ensure clarity and appropriateness for the study population.

Students were informed about the study and invited to complete the self-administered online questionnaire by nursing educators at the end of formal contact sessions. While online surveys tend to offer flexibility whereby participants can complete the questionnaire at their most convenient space and time [30], the response rate is characteristically low [30,31], and hybrid approaches are increasingly advised [32]. Thus, a trained fieldworker distributed hard-copy questionnaires to the students after contact sessions. The free Wi-Fi provided by the university allowed qualifying nursing students to access the online questionnaire via the university’s web-based course-management platform. The platform is designed to enable students and faculty to participate in online academic activities, virtual chats, and interactions. The learning platform is accessible on electronic devices, including desktops, laptops and smartphones. Alternatively, students could complete hard-copy questionnaires. The trained fieldworker handed out hard copy questionnaires and envelopes together with information about the study to nursing students after their lectures. To avoid any potential coercion during recruitment, the faculty were requested to exit the classroom. Respondents were informed that by completing the hard copy questionnaires, they were voluntarily agreeing to participate in the study. Likewise, those who completed the online questionnaire checked a box to indicate voluntary participation in the study. Completed hard-copy questionnaires were enclosed in sealed envelopes and collected as the students filed out of the lecture rooms.

Participation in the study was entirely voluntary. Prior to obtaining consent, nursing students were briefed about the study’s purpose. Consent was implied upon the completion of the survey. Participant anonymity was prioritised, with no names recorded on the self-administered questionnaires. However, to prevent duplicate survey submissions and ensure equitable distribution of gift vouchers, student registration, identification, and cell phone numbers were collected on a separate (verification) form for administrative purposes only.

Privacy and confidentiality were maintained by storing identifying information separately from participants’ responses. The fieldworker received training to ensure the security of both paper-based and electronic data. Electronic records were stored on a password-protected computer, while hard copies were securely stored in locked cabinets in the researchers’ offices, minimising the risk of unauthorised access. Access to the students’ personal information and data was restricted to the research team.

Although no emotional distress was anticipated, the research team acknowledged that discussions surrounding mental health could potentially evoke personal or cultural discomfort due to associated stigma. To address this, respondents were informed of available support resources, including the 24 h helplines provided by the South African Depression and Anxiety Group and the university’s student counselling services, should they experience any emotional distress. Participation in the study was cost free, and to express appreciation, each student received a ZAR 100 (approximately USD 5) gift voucher.

### 2.5. Data Analysis

Unique numbers were used to verify and reconcile online and hard-copy questionnaires to prevent duplication. The data were cleaned and analysed in SPSS 29. Data were described using frequency counts and percentages for categorical variables and measures of central tendency (mean, standard deviation, median, inter-quartile range). Composite scores were also computed for the scale-based variables. The reliability (internal consistency) of the standardised scales was determined by calculating Cronbach’s alpha coefficients. T-tests and one-way analysis of variance (ANOVA) were performed to assess differences (MHLS) across year levels, followed by Tukey’s HSD post hoc tests for pairwise comparisons. An additional one-way ANOVA grouped students by level of mental health exposure (none, theoretical only, and both theoretical and clinical). Multiple linear regression analysis was performed to determine the factors associated with nursing students’ MHL. Statistical significance was determined at *p* ≤ 0.05 and a 95% confidence interval (CI). Preliminary analyses were performed to ensure that the assumptions of linearity, independence of errors, homoscedasticity, unusual points, and normality of residuals were met.

### 2.6. Ethical Consideration

Ethical clearance for the study was obtained from an institutional research ethics committee accredited by the National Health Research Ethics Council of South Africa. Furthermore, formal permission to conduct the study was granted by the university.

## 3. Results

### 3.1. Sociodemographic Characteristics

Table 1 illustrates the respondents’ demographic profile. Of the 220 respondents, the majority were female (*n* = 196, 89%), and the median age (IQR) was 20 years (19–22). The response rate among final year students was relatively low, comprising only 9% (*n* = 20) in comparison to other years of study. The mean GHQ-28 score was 29.2 (SD: 18.5), suggesting that students experienced a moderate level of psychological distress. Self-reported exposure to a mental health condition varied, with 16% (*n* = 35) self-reporting personal experience, 26% (*n* = 58) related to family history and 39% (*n* = 86) linking it to peer illness. Only a third (*n* = 73) of the students indicated that they would specialise in mental health care.

### 3.2. Mental Health Literacy Scale Scores

A one-way ANOVA analysis revealed a statistically significant difference in MHLS scores across nursing student year levels, F(3, 216) = 3.225, *p* = 0.023. Descriptive statistics showed a progressive increase in mean MHLS scores from first-year (mean = 124.50; SD = 10.70), second-year (mean = 126.90; SD = 11.29), third-year (mean = 128.90; SD = 9.72), to fourth-year (mean = 131.75; SD = 12.80). Post hoc comparisons using the Tukey HSD test indicated that the only statistically significant difference was between first- and fourth-year students (*p* = 0.042). All other pairwise comparisons were not statistically significant (all *p* > 0.05). To further explore the role of mental health exposure type, a second one-way ANOVA was conducted by grouping students into three categories: first-year students (no exposure), second-year students (theoretical exposure), and combined third-/fourth-year students (theoretical and clinical exposure). This analysis also revealed a statistically significant difference, F(2, 217) = 4.32, *p* = 0.015. Tukey post hoc results showed that MHLS scores were significantly higher among third-/fourth-year students compared with first-year students (*p* = 0.01), but no significant differences were found between second-year students and either of the other two groups (first-years, *p* = 0.392; third-/fourth-years, *p* = 0.295).

Regarding the six attributes of the MLHS, the total mean score was 127.01 (SD: 10.99) with mean scores ranging from a minimum of 74 to a maximum score of 151 (Table 2). An independent samples *t*-test further revealed that senior nursing students’ scores on the attribute the ability to recognise disorders sub-scale were not statistically different compared with the first-year students (t = −1.500, df = 218, *p* = 0.14). Similarly, senior nursing students’ scores were not statistically significantly different from first-year students’ scores on the knowledge of risk factors and causes of mental illness attribute (mean = 5.1; SD = 1.29 vs. mean = 5.08; SD = 1.17; t = −1.811, df = 218, *p* = 0.072), the knowledge of self-treatment scores (mean = 5.6; SD = 1.15 vs. mean = 5.69; SD = 0.92; t = 0.696, df = 218, *p* = 0.487), knowledge of professional help sub-scale (mean = 9.68; SD = 1.43 vs. mean = 9.91; SD = 1.49; t = 1.163, df = 218, *p* = 0.246), and the attitudes that promote recognition or appropriate help-seeking behaviour (stigma) sub-scale (mean = 65.37; SD = 8.50 vs. mean = 63.34; SD = 7.73; t = −1.726, df = 218, *p* = 0.086). Conversely, senior nursing students’ scores on knowledge of how to seek information sub-scale were statistically significantly higher (mean = 15.38; SD = 2.81 vs. mean = 14.12; SD = 2.89) than first-year students’ scores (t = −3.112, df = 218, *p* = 0.002), an effect size 0.44 as measured by Cohen’s *d*, indicating small to medium effect size.

The mean (SD) MHLS score: 127.1 (SD: 10.98). A closer analysis of the MHLS items revealed variability in responses across the 4-point Likert scale items, with mean scores ranging from 1.13 to 3.71 (minimum = 1, maximum = 4) (Table 3). The item “To what extent do you think it would be helpful for someone to ‘avoid all activities or situations that made them feel anxious if they were having difficulties managing their emotions?” received the lowest mean score of 1.13 (SD: 1.39). This item, which was reverse scored, suggests a low level of agreement with the statement, indicating that respondents generally disagreed with avoidance as a coping strategy. In contrast, the highest mean score was observed for the item “To what extent do you think it is likely that the diagnosis of ‘bipolar disorder’ includes experiencing periods of elevated (i.e., high) and periods of depressed (i.e., low) mood,” with a mean of 3.71, indicating strong consensus and understanding of this diagnostic feature.

Regarding the items scored on a 5-point Likert scale, mean scores ranged from 2.47 to 4.66 (minimum = 1 and maximum = 5). The lowest mean score was observed for the item “I am confident attending face-to-face appointments to seek information about mental illness [e.g., seeing the GP]” 2.47 (SD: 1.39), indicating relatively low confidence among the participants in attending in-person consultations. Conversely, the highest mean score was observed for the reverse-scored item “Seeing a mental health professional means you are not strong enough to manage your own difficulties”, suggesting strong disagreement with the notion that seeking help reflects a lack of personal strength.

### 3.3. Factors Associated with MHL

A multiple linear regression analysis was conducted to examine factors associated with MHL among the student nurses, considering gender, year of study, self-reported general health, personal and family history of mental illness, knowledge of a close friend with mental illness, and interest in specialising in mental healthcare (Table 4). A sample of at least 103 respondents was required to achieve 80% statistical power, accounting for seven predictor variables and a medium (0.15) effect size. All the assumptions necessary for the multiple linear regression analysis—linearity, independence of errors, homoscedasticity, no influential outliers and normality of residuals—were met.

The combined predictor variables significantly predicted MHL among student nurses (F(7, 210) = 7.308, *p* < 0.001, adjusted R^2^ = 0.169). Approximately 16.9% of the variance in MHL (as measured by the MHLS) among nursing students can be explained by the combined predictor variables in the model. Among these variables, year of study (*p* = 0.015), family history of mental illness (*p* = 0.021), and whether respondents would specialise in mental healthcare (*p* < 0.001) showed significant associations with MHL. While gender initially appeared significant (*p* = 0.039), a bootstrap sensitivity analysis suggested that this result may not be robust (bootstrap *p* = 0.170), likely due to the small number of male participants (*n* = 22) in the study. Specifically, female students scored 4.8 points higher on the MHLS compared with male students, but this estimate should be interpreted with caution. Senior students scored 3.7 points higher on the MHLS than first-year students. Students with a family member diagnosed with mental illness scored 4.0 times higher on the MHLS than those who did not have such family members. Students interested in specialising in mental healthcare scored 5.5 points higher than those without such interest.

## 4. Discussion

This study contributes to the expanding literature on the characteristics of and factors associated with MHL among undergraduate nursing students in South Africa. This study observed overall year-level differences in MHL scores, with senior students in higher years of nursing training demonstrating higher scores compared to their first-year counterparts with no exposure to mental health training. These results are comparable to those reported in studies among nursing students and health care workers in South Africa, Zambia, and Turkey [28,33]. However, an unexpected finding in this study was the absence of statistically significant differences in MHL between second-year nursing students (with theoretical exposure only) and those in their final years (with both theoretical and clinical exposure). Implicitly, clinical exposure in this study contributed to limited improvement in MHL. This finding may be attributed to inadequate supervision, limited duration, absence of role modeling and suboptimal integration of theory into practice affecting the quality of clinical placements [7,8,9,10]. These findings mirror similar concerns reported in Iran [34] and the United Kingdom [8] and highlight a potential broader shortcoming within the undergraduate nursing programme—mental health education may be insufficiently developed and inadequately delivered. Perhaps, the SANC, as the regulatory body for nursing training, should review the curriculum—theoretical content and clinical exposure—and teaching strategies to ensure that MHL is being effectively and progressively reinforced throughout the nursing training programme.

In the current study, nursing students strongly disagreed with avoidance as a coping strategy for anxiety. This finding is significant as it implies that students, to a certain extent, demonstrated an understanding that avoidance behaviours are a core feature of anxiety disorders, often responsible for perpetuating the condition [35]. Importantly, a concerning finding to emerge from this study is that despite positive attitudes towards professional help-seeking, students displayed a reduced likelihood to seek treatment for a mental health illness through face-to-face consultations. More research should be undertaken to understand barriers associated with actual in-person evaluations for mental health disorders. Additionally, future research should explore whether alternatives to face-to-face consultations and other forms of remote consultations, such as telehealth for mental health conditions that do not necessitate physical examination, can increase the likelihood of attending face-to-face consultations among nursing students.

Apart from academic seniority, multiple regression analysis initially revealed that gender, family mental history, and professional aspirations were also important factors in predicting MHL among nursing students. However, a bootstrap sensitivity analysis indicated that the gender effect may not be statistically robust, likely due to the small number of male participants in the sample. Despite this, the observed trend—where female students scored higher across several MHLS sub-scales than their male counterparts—aligns with evidence from previous studies elsewhere [36,37,38]. This pattern contributes to the broader understanding of gender differences in MHL. Previous research suggests that women traditionally assume caregiving roles [39], which may increase their engagement with health information. As a result, females have greater familiarity in navigating mental health care services and are more likely to seek help [36,37]. In contrast, men tend to have significantly more stigmatising mental health attitudes [40], limited ability to recognise symptoms of a mental illness [36,41], and a reduced willingness to seek care [36]. It is conceivable, therefore, that these contrasting tendencies are likely causes for the existing gap in MHL between female and male students in the present study. Future research could explore gender-specific targeted interventions to reduce the disparities in MHL among nursing students.

In contrast to earlier research [42,43,44], the present study yielded mixed results regarding the association between exposure to mental health illness and MHL. Specifically, a family history of mental health illness was associated with high MHL levels, suggesting that increased familiarity within the family context may enhance the likelihood of improving MHL. These results are consistent with those from a study conducted in Turkey [33]. In that study, nursing students with a family member with a mental health illness demonstrated reduced stigmatisation and greater empathy and were better prepared to cope and navigate the complexities of mental health challenges. Similarly, in an Australian study, individuals with a family history of mental health illness had increased familiarity with symptoms and a higher likelihood of recognising treatment options [44].

However, neither personal history of a mental health illness nor exposure through close friend experiences demonstrated any significant influence on MHL in this study. This is despite almost 40% (*n* = 80) of respondents indicating that they had a close friend who had previously experienced a mental health issue. A recent study in Hong Kong suggested that individuals with close ties to family or friends with mental health illness showed less stigma, while personal experience paradoxically increased self-stigma and reinforced negative attitudes [42]. Undoubtedly, there are multiple interacting factors that contribute to the complexity of this relationship. This rather contradictory result suggests that different exposures to and familiarity with mental illness may influence MHL in distinct ways. Sadly, the mechanisms and the extent to which different exposures to mental health illness influence MHL have seldom been interrogated in depth. Collectively, the findings call for a deeper reexamination of the underlying distinct mechanisms by which different exposures to and familiarity with mental health illnesses influence MHL.

Reinforcing previous literature [9,45,46,47], this study confirmed that mental health care nursing remains an unpopular career choice, with only a third of the students expressing willingness to specialise in the discipline. Earlier studies conducted in South Africa [45] and internationally [9,46,47] have consistently reported a reluctance among health professionals to pursue careers in psychiatry. The most frequently cited reasons include inadequate exposure during training, societal stigma against mental health disorders, and perceived lower prestige. Apart from that, in South Africa, for example, research has shown that nursing students often enter the nursing programme holding cultural beliefs such as attributing mental health disorders to witchcraft or spiritual punishment that can influence their engagement with mental health content and career interest [48,49]. As such, the lack of interest in pursuing a career in mental health suggests a potentially limited pipeline of future specialised mental health practitioners—at a time when they are urgently needed to address the ongoing shortage of human resources for mental health. To shift perceptions and cultivate greater interest in the field, strategic interventions may be required. These could include increasing culturally responsive mental health content within nursing curricula, providing early and positive clinical exposure to psychiatric settings, offering mentorship from mental health professionals, and elevating the status of the discipline within the profession within nursing and the broader health system.

Importantly, the present study raises the possibility that the timing of the introduction to the mental health theory content may have influenced the limited career preference in mental health among nursing students. In this study setting, nursing students were only introduced to mental health theory in the second year, after the foundational year—by which stage students may have already developed misconceptions about the specialty. Previous research has demonstrated that the negative attitudes towards a career in mental health nursing by undergraduate nursing students can be mitigated by adequate educational preparation—through timely exposure to theoretical classroom-based instruction and extended clinical placements [7,9,47,50]. A scaffold curriculum approach that introduces foundational mental health topics early in the first year and progressively expands in complexity and relevance in clinical practice each year should be considered to enhance meaningful MHL gains throughout the nursing training programme. Future research should examine whether an early introduction to mental health nursing content in the formative stages, with incremental increases in clinical training across the years of study, may influence nursing students’ desire to consider a career in mental health nursing.

While the current study highlights important factors associated with MHL among nursing students, there are some limitations that should be considered. This study used a convenience sampling approach targeting all undergraduate nursing students enrolled at the tertiary education institution in 2023. Although 300 students were eligible to participate, only 220 completed the survey, and final-year students were notably underrepresented. The lack of a pilot study may limit the internal validity of the results. However, this limitation was partially addressed by using a previously validated instrument for a similar context and tailored for South African PHC workers. Despite hosting the questionnaire on the course online platform for eight weeks to maximise accessibility, the response rate from this cohort remained low. Consequently, the results may not fully capture the perspectives of students at different stages of training, particularly those nearing programme completion. Moreover, given that the results reflect the MHL of undergraduate nursing students from one South African tertiary institution based on a non-random sampling strategy, the generalisability of results to nursing students in other universities and provinces is limited.

The cross-sectional design of this study precludes the determination of a cause-and-effect relationship between mental health nursing education and self-reported MHL in this setting. Future studies should consider applying a longitudinal pretest–post-test design to more accurately measure the change in MHL throughout undergraduate nursing training. Moreover, this study relied on self-reported MHL responses, with no way to validate the actual behaviours of nursing students in practice. Future research should consider administering a formal measure for social desirability bias, such as the Balanced Inventory of Desirable Responding (BIDR), to account for socially desirable responding. Nevertheless, the potential effect of social desirability bias inherent in self-reports was reduced by assuring the respondents of the confidentiality of their responses. Despite this limitation, the findings contribute important insights into MHL among nursing students and the factors associated with MHL.

## 5. Conclusions

Overall, practical exposure in final years (third-/fourth-years) offered no significant MHL advantage over second-years with theory only, suggesting a broader shortcoming within the undergraduate nursing programme. Implicitly, students were arguably inadequately prepared for their future roles as non-specialist mental health providers and were reluctant to pursue a career in mental health. The findings raise questions about the timing, quality, design, and content of the nursing mental health training curriculum. These findings highlight important insights for stakeholders involved in nursing education, such as the SANC, institutions of higher education, and the Department of Health, to consider in reviewing the curriculum—theoretical content and clinical exposure—and teaching strategies to ensure that MHL is being effectively and progressively reinforced throughout the training programme.

This study found gender, year of study, family history of mental illness, and desire to specialise in mental healthcare to be strong predictors of MHL among nursing students. These study results not only deepen the current understanding of MHL among nursing students but also provide a foundation for future research to investigate the influence of additional individual and contextual factors, such as perceptions of social support and exposure to media campaigns, on mental health and mental health information seeking among nursing students that may influence students’ MHL.

## Figures and Tables

**Table 1 nursrep-15-00337-t001:** Sociodemographic characteristics of the sample (*n* = 220).

Variable	*n* (%)
Gender	
Male	22 (10)
Female	196 (89.1)
Non-binary	2 (0.9)
Age in years (median [IQR]) *	20 (19–22)
Year of study	
First	74 (33.6)
Second	65 (29.6)
Third	61 (27.7)
Fourth	20 (9.1)
General health score (mean, SD)	29.2 (18.5)
Self-reported history of mental illness	
Yes	35 (15.9)
No	180 (81.8)
Prefer not to answer	18 (2.3)
Know a family member with a mental health problem	
Yes	58 (26.4)
No	141 (64.1)
Unsure	21 (9.5)
Know a friend with a mental health problem	
Yes	86 (39.1)
No	116 (52.7)
Unsure	18 (8.2)
Would specialise in mental health care	
Yes	73 (33.2)
No	76 (34.2)
Unsure	71 (32.2)
Mental health literacy (MHL) score (mean, SD)	127 (11.0)

* IQR: Inter-quartile range; SD: standard deviation.

**Table 2 nursrep-15-00337-t002:** Mental Health Literacy Scale attributes.

Attribute	Possible Scores		Participant Scores		
	Minimum	Maximum	Mean (SD)	Minimum	Maximum
The ability to recognise disorders (Q1–Q8)	8	32	26.8 (2.98)	13	32
Knowledge of how to seek information (Q16–Q19)	4	20	14.9 (2.90)	6	20
Knowledge of risk factors and causes of mental illness (Q9, Q10)	2	8	5.30 (1.26)	2	8
Knowledge of self-treatment (Q11, Q12)	2	8	5.62 (1.08)	2	8
Knowledge of professional help available (Q13–Q15)	3	12	6.15 (1.13)	3	8
Attitudes that promote recognition or appropriate help-seeking behaviour (stigma) (Q20–Q35)	16	80	64.69 (8.28)	22	80
Total score (Q1–Q35): minimum = 35; maximum = 160			127.01 (10.99)	74	151

**Table 3 nursrep-15-00337-t003:** Nursing students’ mental health literacy scores.

Item	Mean (SD)	Minimum	Maximum
If someone became extremely nervous or anxious in one or more situations with other people (e.g., a social gathering) or performance situations (e.g., presenting at a meeting) in which they were afraid of being evaluated by others and that they would act in a way that was humiliating or feel embarrassed, then to what extent do you think it is likely they have Social Phobia?	3.38 (0.60)	1	4
2. If someone experienced excessive worry about a number of events or activities where this level of concern was not warranted, had difficulty controlling this worry and had physical symptoms such as having tense muscles and feeling fatigued, then to what extent do you think it is likely they have “Generalised Anxiety Disorder”?	3.43 (0.63)	1	4
3. If someone experienced a low mood for two or more weeks, had a loss of pleasure or interest in their normal activities and experienced changes in their appetite and sleep then to what extent do you think it is likely they have “Major Depressive Disorder”?	3.33 (0.72)	1	4
4. To what extent do you think it is likely that “Personality Disorders” are a category of mental illness?	3.33 (0.77)	1	4
5. To what extent do you think it is likely that “Persistent Depressive disorder” (Dysthymia) is a mental disorder?	2.93 (0.83)	1	4
6. To what extent do you think it is likely that the diagnosis of “Agoraphobia” includes anxiety about situations where escape may be difficult or embarrassing?	3.03 (0.78)	1	4
7. To what extent do you think it is likely that the diagnosis of “Bipolar Disorder” includes experiencing periods of elevated (i.e., high) and periods of depressed (i.e., low) mood	3.71 (0.62)	1	4
8. To what extent do you think it is likely that the diagnosis of “Substance Abuse Disorder” can include physical and psychological tolerance of the drug (i.e., require more of the drug to get the same effect)?	3.63 (0.67)	1	4
9. To what extent do you think it is likely that in general, “women are MORE likely to experience a mental illness of any kind compared to men?”	2.28 (0.90)	1	4
10. To what extent do you think it is likely that in general, “men are MORE likely to experience an anxiety disorder compared to women?” (R)	3.02 (0.93)	1	4
11. To what extent do you think it would be helpful for someone to “improve their quality of sleep” if they were having difficulties managing their emotions (e.g., becoming very anxious or depressed)?	3.38 (0.81)	1	4
12. To what extent do you think it would be helpful for someone to “avoid all activities or situations that made them feel anxious” if they were having difficulties managing their emotions (R)	1.13 (1.39)	1	4
13. To what extent do you think it is likely that “Cognitive Behaviour Therapy (CBT)” is a therapy based on challenging negative thoughts and increasing helpful behaviours?”	3.23 (0.65)	1	4
14. Mental health professionals are bound by confidentiality; however, there are certain conditions under which this does not apply. To what extent do you think it is likely that the following is a condition that would allow a mental health professional to “break confidentiality”: If a patient is at immediate risk of harm to oneself or others?	3.60 (0.72)	1	4
15. Mental health professionals are bound by confidentiality; however, there are certain conditions under which this does not apply. To what extent do you think it is likely that the following is a condition that would allow a mental health professional to “break confidentiality”: if the patient’s problem is not life-threatening and professionals want to assist others to better support a patient (R)	2.92 (1.02)	1	4
16. I am confident that I know where to seek information about mental health illness.	4.05 (0.94)	1	5
17. I am confident using the computer or telephone to seek information about mental illness.	4.13 (1.04)	1	5
18. I am confident attending face-to-face appointments to seek information about mental illness (e.g., seeing the GP).	2.47 (1.39)	1	5
19. I am confident I have access to resources (e.g., GP, internet, friends) that I can use to seek information about mental illness.	4.32(0.99)	1	5
20. People with a mental illness could put themselves together if it if they wanted. (R)	4.05 (1.28)	1	5
21. A mental illness is a sign of personal weakness. (R)	4.63 (0.80)	1	5
22. A mental illness is not a real medical illness. (R)	4.56 (0.90)	1	5
23. People with a mental illness are dangerous. (R)	3.39 (1.09)	1	5
24. It is best to avoid people with a mental illness so that you don’t develop this problem. (R)	4.64 (0.72)	1	5
25. If I had a mental illness I would tell no one. (R)	4.08 (1.05)	1	5
26. Seeing a mental health professional means you are not strong enough to manage your own difficulties. (R)	4.66 (0.86)	1	5
27. If I had a mental illness, I would not seek help from a mental health professional. (R)	4.50 (0.92)	1	5
28. I believe treatment for a mental illness, provided by a mental health professional, would not be effective. (R)	4.55 (0.90)	1	5
29. How willing would you be to move next door to someone with a mental illness?	3.52 (1.13)	1	5
30. How willing would you be to spend an evening socialising with someone with a mental illness?	4.07 (0.99)	1	5
31. How willing would you be to make friends with someone with a mental illness?	4.10 (1.00)	1	5
32. How willing would you be to have someone with a mental illness start working closely with you on a job?	3.70 (1.05)	1	5
33. How willing would you be to have someone with a mental illness marry into your family?	3.55 (1.13)	1	5
34. How willing would you be to vote for a politician if you knew they had suffered a mental illness?	3.03 (1.24)	1	5
35. How willing would you be to employ someone if you knew they had a mental illness?	3.67 (1.13)	1	5

**Table 4 nursrep-15-00337-t004:** Factors associated with MHL among nursing students.

Variable	Unstandardised Coefficients		*p*-Value	Bootstrap *p*-Value
	**B**	**Standard Error**		
Sex				
Male (ref)				
Female	4.795	2.314	0.039	0.170
Year of study				
First (ref)				
Second and higher	3.683	1.501	0.015	0.013
General health score	0.015	0.041	0.712	0.726
Personal history of mental health illness				
No (ref)				
Yes	3.628	2.061	0.080	0.103
Family member diagnosed with mental health illness				
No (ref)				
Yes	3.954	1.705	0.012	0.029
Close friend diagnosed with mental health illness				
No (ref)				
Yes	2.100	1.529	0.171	0.216
Would specialise as a mental health nurse				
No (ref)				
Yes	5.460	1.476	<0.001	<0.001
Constant	115,665	2.764	<0.001	<0.001

## Data Availability

All relevant data are within the manuscript. Additional data is available on request from the corresponding author.

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
