# Peer review of "Assessing Mental Health Literacy Among Undergraduate Nursing Students in South Africa"

_nursrep, 2025, doi:10.3390/nursrep15090337_

Round 1
Reviewer 1 Report
Comments and Suggestions for Authors
Dear authors,
I appreciate the content and quality of this manuscript. Thank-you for this work. My suggestions would be :
1) I think it is clunky to say mental health illness. I do understand that is now how we refer to this but it seems odd, perhaps mental health challenge (s)?
2) Perhaps in the discussion the authors could offer another strategy? Another way to increase mental health literacy and decrease stigma would be to intentionally integrate mental health theory and concepts throughout the curriculum, including in the first year. Start with what does mental health look like and slowly introduce basic concepts (stress, anxiety, depression, self-care, etc) and help students assess for and look at how mental health is part of all our lives.
3) I would also say that it may be helpful to include a few sentences on the impact of the cultural context perspectives on mental illness.
Author Response
Comment 1: I appreciate the content and quality of this manuscript. Thank-you for this work.
Response 1: We would like to thank the reviewer for carefully and thoroughly reading our manuscript and for the thoughtful and encouraging comments.
Comment 2: I think it is clunky to say mental health illness. I do understand that is now how we refer to this but it seems odd, perhaps mental health challenge (s)?
Response 2: We appreciate this observation. We note that the Mental Health Literacy Scale (MHLS) used in our study, as adapted by Korhonen et al. (2022) for primary health care workers in South Africa and Zambia, includes questions that uses the phrase ‘mental health illness’. This influenced the use of the terminology mental health illness that we retained throughout in core reporting to maintain consistency with the original tool and ensure measurement precision. However, drawing from the excellent point you raised, we have selectively replaced it with ‘mental health challenge(s)’ or ‘disorder(s)’ in the discussion and narrative sections where such changes does not compromise the accuracy of our messaging and alignment with the tool’s original content.
Comment 3: Perhaps in the discussion the authors could offer another strategy? Another way to increase mental health literacy and decrease stigma would be to intentionally integrate mental health theory and concepts throughout the curriculum, including in the first year. Start with what does mental health look like and slowly introduce basic concepts (stress, anxiety, depression, self-care, etc) and help students assess for and look at how mental health is part of all our lives.
Response 3: Thank you for this excellent suggestion. We have revised the discussion section recommending a longitudinal, scaffolded approach to curriculum design. This includes introducing basic mental health concepts from the first year and progressively building mental health literacy throughout the nursing programme [line 461 – 464].
Comment 4: I would also say that it may be helpful to include a few sentences on the impact of the cultural context perspectives on mental illness
Response 4: We agree. We have incorporated additional discussion on how sociocultural perspectives within the South African context may influence student attitudes toward mental health and a career in the field [line 439 – 443, 448].
Reviewer 2 Report
Comments and Suggestions for Authors
This is a well-designed study with an important analysis about mental health literacy that could help to have a perspective about students in the changes of theory and clinical practice due to the necessity of psychiatric nurses. This topic is relevant and could be examined for other investigations that would consider mental health literacy and how they can impact understanding of mental diseases for different careers and possibly in different contexts. It could be relevant if this study could be replicated with different populations in progressive year-levels to have a more profound understanding for improving mental health services.
The experimental design is adequate for the proposed analysis, studying a nursing program to observe how different changes in a four-year programme can affect mental health literacy. Statistical results presentation is accurate for the results showing a low improvement in senior students compared with other groups. Your results are well discussed with previous work. I recommend acceptance with minor corrections.
Methods. It is necessary to present a general syllabus of each year course of mental health theory for the different years to observe if the association is due to the intrinsic considerations of the topic or is the design of the programme.
L317. In the Item “I am confident attending face-to-face appointments to seek information about mental illness [e.g., seeing the GP]”4.32 (SD: 0.99)”, it seems to be a mistake that could correspond to the highest mean score not the lowest and with the writing "seek information about information".
Author Response
Comment 1: This is a well-designed study with an important analysis about mental health literacy that could help to have a perspective about students in the changes of theory and clinical practice due to the necessity of psychiatric nurses. This topic is relevant and could be examined for other investigations that would consider mental health literacy and how they can impact understanding of mental diseases for different careers and possibly in different contexts. It could be relevant if this study could be replicated with different populations in progressive year-levels to have a more profound understanding for improving mental health services.
The experimental design is adequate for the proposed analysis, studying a nursing program to observe how different changes in a four-year programme can affect mental health literacy. Statistical results presentation is accurate for the results showing a low improvement in senior students compared with other groups. Your results are well discussed with previous work. I recommend acceptance with minor corrections.
Response 1: Thank you for this encouraging feedback. We agree that replication in other contexts and across year levels would yield deeper insights and help improve mental health education for broader health professions.
Comment 2: Methods. It is necessary to present a general syllabus of each year course of mental health theory for the different years to observe if the association is due to the intrinsic considerations of the topic or is the design of the programme.
Response 2: Thank you for the suggestion. We have now added a brief overview of the structure and focus of mental health content across the years of the nursing programme [line 131 – 133].
Comment 3: L317. In the Item “I am confident attending face-to-face appointments to seek information about mental illness [e.g., seeing the GP]”4.32 (SD: 0.99)”, it seems to be a mistake that could correspond to the highest mean score not the lowest and with the writing "seek information about information".
Response 3: Thank you for spotting this. We have corrected the typographical error and clarified that this item recorded as the lowest mean score. The revised sentence now reads correctly as follows:
The lowest mean score was observed for the item “I am confident attending face-to-face appointments to seek information about mental illness [e.g., seeing the GP]” 2.47 (SD: 1.39), indicating relatively low confidence among the participants in attending in-person consultations [line 323 – 324].
Reviewer 3 Report
Comments and Suggestions for Authors
A well refined work for explanation and for its methodological solidity. Even if the sample is locally defined is sufficiently generalizable. It could be used some other af instruments as BIDR for social desirability to control this confounding variable
Author Response
Comment 1: A well refined work for explanation and for its methodological solidity. Even if the sample is locally defined is sufficiently generalizable. It could be used some other af instruments as BIDR for social desirability to control this confounding variable.
Response 1: Thank you for this useful suggestion. While we did not include a formal measure for social desirability bias such as the BIDR, we acknowledge its potential influence in self-reported data. We have noted this limitation in the Discussion and suggested it for future studies [line 492 – 495].
Reviewer 4 Report
Comments and Suggestions for Authors
- Introduction: The background is comprehensive and supported by appropriate literature. However, I would suggest integrating more literature on MHL trends outside of South Africa to contextualize the findings internationally.
- Methodology: Due to absence of a pilot study, as stated by the authors in line 181, I would suggest further discussing the possible implications on validity.
- Discussion: A more nuanced reflection on possible reasons why clinical exposure did not translate into MHL gains (lines 359 - 362) would be beneficial (e.g. inadequate supervision and mentorship, negative role-modeling, student disengagement, lack of interest, duration and variety of placements, etc.)
Author Response
Comment 1: Introduction: The background is comprehensive and supported by appropriate literature. However, I would suggest integrating more literature on MHL trends outside of South Africa to contextualize the findings internationally.
Response 1: We appreciate this suggestion and have expanded the Introduction to include comparative trends and insights from high income countries, including Australia [line 92 – 94].
Comment 2: Methodology: Due to absence of a pilot study, as stated by the authors in line 181, I would suggest further discussing the possible implications on validity.
Response 2: Thank you. We have added a reflection on the potential implications for instrument validity in the Limitations section. We also noted that we used a previously validated and context-adapted tool, which partially mitigates this limitation [line 476 – 478].
Comment 3: Discussion: A more nuanced reflection on possible reasons why clinical exposure did not translate into MHL gains (lines 359 - 362) would be beneficial (e.g. inadequate supervision and mentorship, negative role-modelling, student disengagement, lack of interest, duration and variety of placements, etc.).
Response 3: Thank you. We have expanded this section in the Discussion to include possible reasons such as inadequate supervision, limited duration, absence of role modelling and suboptimal integration of theory into practice [line 368 – 370].